# Alignment Precedes Fusion: Open-Vocabulary Named Entity Recognition as Context-Type Semantic Matching

**Zhuoran Jin[1,2], Pengfei Cao[1,2], Zhitao He[1,2], Yubo Chen[1,2], Kang Liu[1,2], Jun Zhao[1,2]**

[1] School of Artificial Intelligence, University of Chinese Academy of Sciences, Beijing, China
[2] The Laboratory of Cognition and Decision Intelligence for Complex Systems,
Institute of Automation, Chinese Academy of Sciences, Beijing, China
{zhuoran.jin, pengfei.cao, zhitao.he}@nlpr.ia.ac.cn
{yubo.chen, kliu, jzhao}@nlpr.ia.ac.cn

## Abstract

Despite the significant progress in developing named entity recognition models, scaling to novel-emerging types still remains challenging in real-world scenarios. Continual learning and zero-shot learning approaches have been explored to handle novel-emerging types with less human supervision, but they have not been as successfully adopted as supervised approaches. Meanwhile, humans possess a much larger vocabulary size than these approaches and have the ability to learn the alignment between entities and concepts effortlessly through natural supervision. In this paper, we consider a more realistic and challenging setting called open-vocabulary named entity recognition (OVNER) to imitate human-level ability. OVNER aims to recognize entities in novel types by their textual names or descriptions. Specifically, we formulate OVNER as a semantic matching task and propose a novel and scalable two-stage method called **C**ontext-Type Sem**A**nti**C A**lignment and Fusi**O**n (**CACAO**). In the pre-training stage, we adopt Dual-Encoder for context-type semantic alignment and pre-train Dual-Encoder on 80M context-type pairs which are easily accessible through natural supervision. In the fine-tuning stage, we use Cross-Encoder for context-type semantic fusion and fine-tune Cross-Encoder on base types with human supervision. Experimental results show that our method outperforms the previous state-of-the-art methods on three challenging OVNER benchmarks by 9.7%, 9.5%, and 1.8% F1-score of novel types. Moreover, CACAO also demonstrates its flexible transfer ability in cross-domain NER. [1]

## 1 Introduction

Named entity recognition (NER) is a fundamental task in natural language processing, focusing on locating named entities mentioned in unstructured text and classifying them into pre-defined types,

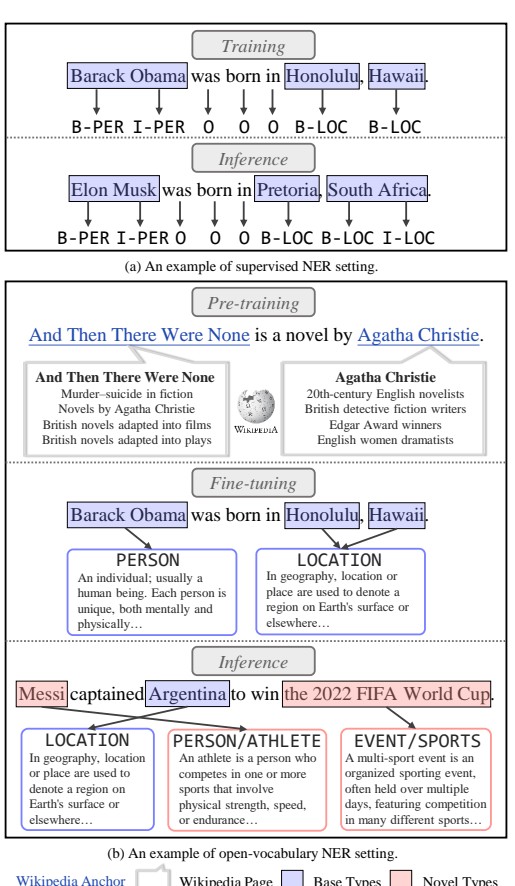

Figure 1: Difference between (a) supervised NER setting and (b) open-vocabulary NER setting.

such as PERSON, LOCATION, ORGANIZATION, etc. Despite its challenging nature, recent methods have achieved high performance on several supervised benchmarks like CoNLL03 (Tjong Kim Sang and De Meulder, 2003) and OntoNotes 5.0 (Pradhan et al., 2013). However, a practical challenge remains in the real-world applications of NER. As depicted in Figure 1 (a), in the supervised setting, traditional methods depend on the pre-defined entity ontology and limited annotated training data. Therefore, these methods can only recognize a fixed number of entity types observed during the training stage. As novel entity types continually

---

[1] Our code and data is publicly available at https://github.com/jinzhuoran/OVNER/

emerge in real scenarios, it becomes a non-trivial problem for NER models to handle the **novel-emerging** types on the fly at inference time.

To solve the challenge of novel-emerging types, a straightforward approach is to collect more training examples for new types and then re-train the model from scratch. The large cost of data collection and model re-training limits the usage of this approach. To tackle this problem, Cao et al. (2020) and Monaikul et al. (2021) propose continual learning approaches for sequence labeling, which require the model to learn new types incrementally without forgetting previous types. Continual learning approaches are inefficient due to the requirement of collecting enough annotated data for new types. Wang et al. (2022) propose continual few-shot NER to reduce the need for annotated data. Hence, Aly et al. (2021) propose a more efficient zero-shot learning approach for NER, which can recognize unseen entity types without any annotated data. However, zero-shot learning methods learn a limited set of base types and struggle to generalize to novel types in practical applications. There is a large performance gap between the current zero-shot NER and supervised NER models. Thus, existing zero-shot NER is unpractical.

Compared to the limited types of the above approaches, humans use thousands of vocabulary to describe the entity types. In cognitive psychology, humans can learn to connect visual objects and semantic categories effortlessly through natural supervision, so they have the ability to recognize unseen categories with a wide-ranging vocabulary (Lake et al., 2017; Zareian et al., 2021). Inspired by this, we argue that humans can expand their vocabulary by reading millions of articles, and learn the alignment between entities and concepts through natural supervision. As illustrated in Figure 1 (b), take the sentence "*And Then There Were None is a novel by Christie.*" in a Wikipedia article as an example. We can connect entity context "*Christie*" and descriptive concepts (e.g., "*British detective fiction writers*") of the entity "*Agatha Christie*" through the anchor (i.e., hyperlink) naturally.

To imitate human-level ability, this work considers a more realistic and challenging setting called open-vocabulary named entity recognition (OVNER). Open-vocabulary setting requires that the model can recognize entities in any novel type by their textual names or descriptions. It is an essential step toward reducing laborious human

supervision and facilitating human-computer interaction. As shown in Figure 1 (b), we devise a two-stage framework for OVNER. In the pre-training stage, based on Wikipedia articles and anchors, we first construct a large-scale pre-training corpus called Wikipair80M. Wikipair80M contains 80M context-type pairs which are freely available on Wikipedia. Then, we pre-train the recognizer to expand its vocabulary, simulating human learning by natural supervision. In the fine-tuning stage, we further fine-tune the recognizer on base types (e.g., LOCATION) with costly annotations to realize open-vocabulary recognition. These two stages enable the recognizer to generalize novel types (e.g., EVENT/SPORTS) without any annotation.

More specifically, we formulate OVNER as a semantic matching task and propose a novel two-stage method called **C**ontext-Type Sem**A**nti**C** **A**lignment and Fusi**O**n (**CACAO**). In the pre-training stage, we adopt Dual-Encoder (two encoders) architecture for efficient training on large-scale, easily accessible context-type pairs. For context-type semantic alignment, Dual-Encoder embeds entity context and type descriptions into a universal dense space independently via in-instance and in-batch contrastive learning. In the fine-tuning stage, we employ Cross-Encoder (single encoder) architecture for effective tuning on limited NER annotations. For context-type semantic fusion, Cross-Encoder captures the intricate interactions between context and all candidate types based on the representations encoded by Dual-Encoder. Experimental results on three challenging OVNER benchmarks demonstrate that our approach significantly outperforms previous state-of-the-art methods.

Our contributions are summarized as follows:

- To solve the challenge of novel-emerging types, we propose a more realistic and challenging setting called open-vocabulary named entity recognition (OVNER), which aims to recognize entities in any novel type by their names or descriptions.

- To imitate human-level ability, we formulate OVNER as a semantic matching task and propose a two-stage method called **C**ontext-Type Sem**A**nti**C** **A**lignment and Fusi**O**n (**CACAO**), which enables the model to learn from both natural and human supervision.

- To evaluate the effectiveness of CACAO, we conduct thorough experiments and show that

our method outperforms the previous state-of-the-art methods on three challenging benchmarks by 9.7%, 9.5%, and 1.8% F1-score of novel types. CACAO also demonstrates its flexible transfer ability in cross-domain NER.

## 2   Task Definition

In this section, we formulate open-vocabulary named entity recognition task as follows. At the pre-training time, large-scale context-type pairs $(c, t)$ are available. At the fine-tuning time, the model can only access the annotated data of base types $\mathcal{T}_B$ but need to expand the vocabulary to recognize novel types $\mathcal{T}_N$ at test time. Since $\mathcal{T}_B \cap \mathcal{T}_N = \emptyset$, it is very challenging for model to generalize to unseen types. Each type $t \in \mathcal{T}_B \cup \mathcal{T}_N$ has its name and detailed description. Given a sequence of $n$ tokens $x = \{x_1, x_2, \ldots, x_n\}$ as the context $c$, the NER model aims to assign each token $x_i$ to its corresponding tag label $y_i = \{y_i^1, y_i^2, \ldots, y_i^{|\mathcal{T}_B| + |\mathcal{T}_N|}\}$, where $y_i^j \in \{0, 1\}$ denotes whether the $i$-th token belongs to the $j$-th type.

## 3   Methodology

Most prior studies solve supervised NER via the sequence labeling paradigm, which conducts multi-class classification on each token in the sentence. However, the sequence labeling paradigm converts all types into indices directly, ignoring the semantic information of the types. Therefore, we formulate OVNER as a semantic matching task. More specially, we propose a two-stage method called **C**ontext-Type Sem**A**nti**C** **A**lignment and Fusi**O**n (CACAO) to solve OVNER problem, which is illustrated in Figure 2. CACAO consists of context-type semantic alignment pre-training stage (Section 3.1) and open-vocabulary named entity recognition fine-tuning stage (Section 3.2). We will detail the model architectures, score functions, and optimization objectives of these two stages.

### 3.1   Context-Type Semantic Alignment Pre-training

In the context-type semantic alignment pre-training stage, we first construct a large-scale pre-training corpus based on Wikipedia articles and anchors. The corpus contains 80M context-type pairs which are freely available on Wikipedia. Then, we pre-train the recognizer to expand its vocabulary, simulating human learning by natural supervision for a rich and complete context-type semantic space.

**Architecture**   We use Dual-Encoder architecture to model context-type pairs similar to entity linking (Wu et al., 2020; Zhang et al., 2022a,b; Leszczynski et al., 2022). As shown in Figure 2 (a) and (b), Dual-Encoder consists of two independent transformer encoders, called Context-Encoder $E_{\text{ctxt}}$ and Type-Encoder $E_{\text{type}}$. Both input context and type description are encoded into vectors:

$$[\mathbf{h}_{c,1}, \mathbf{h}_{c,2}, ..., \mathbf{h}_{c,n}] = E_{\text{ctxt}}(x_c), \qquad (1)$$

$$\mathbf{h}_t = \text{red}(E_{\text{type}}(x_t)), \qquad (2)$$

where $x_c$ and $x_t$ are token sequences of context $c$ and type $t$ respectively. Specifically, we construct the input of context as:

[CLS] *context* [SEP] ,

where *context* are word-pieces tokens of input context. We construct the input of type as:

[CLS] *name* [SEP] *description* [SEP] ,

where *name* and *description* are word-pieces tokens of type name and description. $E_{\text{ctxt}}$ and $E_{\text{type}}$ are two pre-trained language models (Sun et al., 2022; Zhao et al., 2023), such as BERT (Devlin et al., 2019) and RoBERTa (Liu et al., 2019). $\text{red}(\cdot)$ is a pooling function that transforms multiple vectors into one vector. We experiment with three $\text{red}(\cdot)$ functions: using the output vector of the [CLS] token, using the output vector of the *name* token, and computing the mean of all output vectors.

**Scoring**   The similarity score between the token in context $c$ and the type $t$ is computed as:

$$\text{sim}(\mathbf{h}_c, \mathbf{h}_t) = \frac{\mathbf{h}_c^\top \mathbf{h}_t}{\|\mathbf{h}_c\| \cdot \|\mathbf{h}_t\|}. \qquad (3)$$

**Optimization**   For efficient pre-training, we optimize Dual-Encoder by context-type contrastive representation learning. Contrastive learning (Chen et al., 2020; Gao et al., 2021) aims to learn effective representation by pulling positive examples together and pushing apart negative examples. We argue that (1) the token semantics are determined by its context; (2) the token and its type should have the same semantics. Therefore, we can treat all the context-type pairs as positive pairs $(x_{c,+}, x_t)$, where $x_{c,+}$ denotes the token in the context $c$ belongs to the type $t$. As shown in Figure 2 (a) and (b), we devise two contrastive strategies: (1) **in-instance contrastive pre-training**: we treat all the tokens in the context $c$ except $x_{c,+}$ as negative examples. Take Figure 2 (a) as an example, the

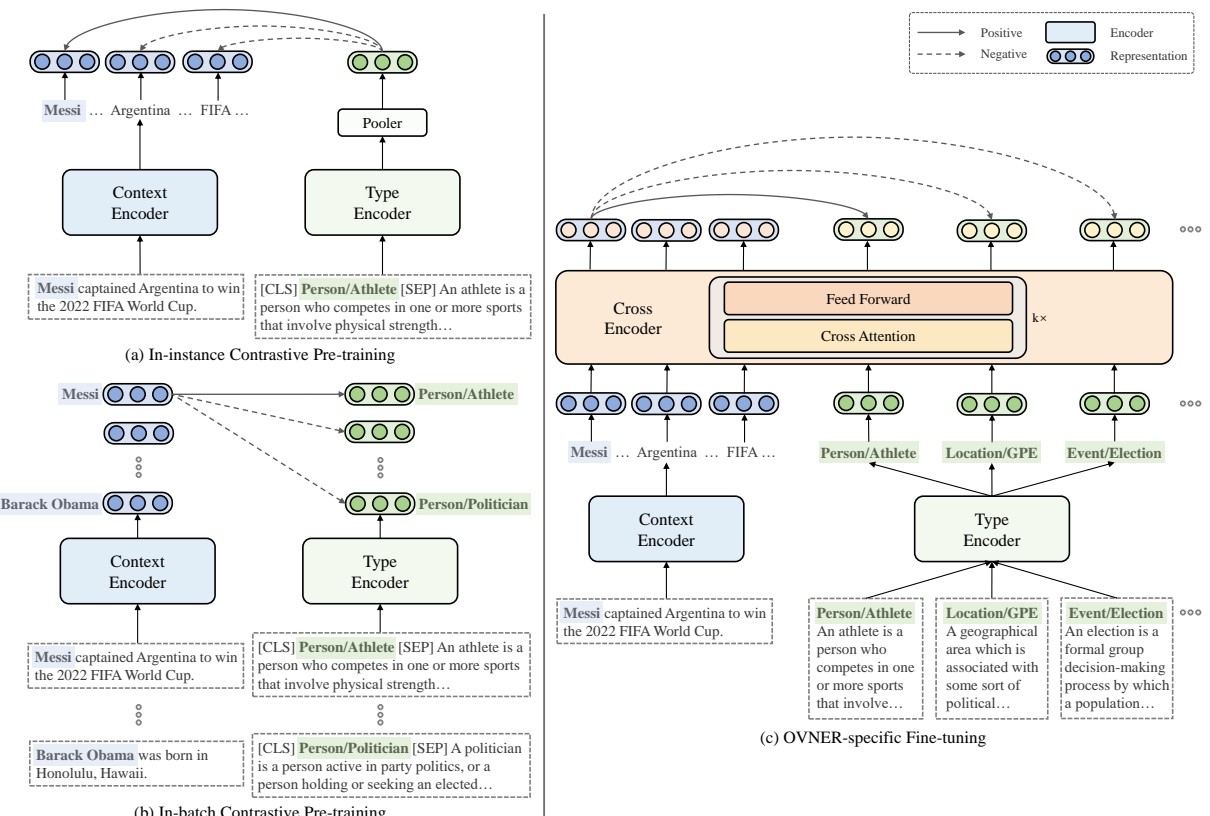

Figure 2: The two-stage framework of our proposed CACAO for open-vocabulary named entity recognition, which consists of context-type semantic alignment pre-training stage (Left Figure (a) and (b)) and open-vocabulary named entity recognition fine-tuning stage (Right Figure (c)).

type `Person/Athlete` is the anchor, the token "*Messi*" is a positive example, and the other tokens in the context "*Messi captained Argentina to win the 2022 FIFA World Cup.*" are negative examples. The training objective of in-instance contrastive pre-training becomes:

$$\mathcal{L}_{inst} = -\log \frac{e^{\text{sim}(\mathbf{h}_t, \mathbf{h}_{c,+})/\tau}}{\sum_{i=1}^{n} e^{\text{sim}(\mathbf{h}_t, \mathbf{h}_{c,i})/\tau}}, \quad (4)$$

where $\tau$ denotes the temperature hyperparameter; (2) **in-batch contrastive pre-training**: we construct the in-batch negative examples similar to Li et al. (2022d). Take the $j$-th instance in a mini-batch of $m$ instances, the training objective of in-batch contrastive pre-training becomes:

$$\mathcal{L}_{t2c} = -\log \frac{e^{\text{sim}(\mathbf{h}_{c,+}^j, \mathbf{h}_t^j)/\tau}}{\sum_{i=1}^{m} e^{\text{sim}(\mathbf{h}_{c,+}^j, \mathbf{h}_t^i)/\tau}}, \quad (5)$$

$$\mathcal{L}_{c2t} = -\log \frac{e^{\text{sim}(\mathbf{h}_t^j, \mathbf{h}_{c,+}^j)/\tau}}{\sum_{i=1}^{m} e^{\text{sim}(\mathbf{h}_t^j, \mathbf{h}_{c,+}^i)/\tau}}, \quad (6)$$

$$\mathcal{L}_{bat} = \frac{1}{2} \left( \mathcal{L}_{t2c} + \mathcal{L}_{c2t} \right). \quad (7)$$

Then, we can get the final optimization goal of the pre-training stage:

$$\mathcal{L}_{pre} = \mu_1 \mathcal{L}_{inst} + \mu_2 \mathcal{L}_{bat}, \quad (8)$$

where $\mu_1$ and $\mu_2$ are hyper-parameters.

To encourage the model to learn entity semantics from context and prevent the model from relying on superficial statistical cues. We randomly mask the entity tokens in context with `[MASK]` token in probability $p$.

**Data Collection** For CACAO pre-training, we need to collect large-scale, easily accessible context-type pairs $(c, t)$. We collect the pre-train corpus through KILT (Petroni et al., 2021), which is based on the 2019/08/01 Wikipedia snapshot and contains 5.9M Wikipedia pages. Each Wikipedia page is assigned: (1) a unique Wikipedia id; (2) a list of paragraphs; (3) a list of anchors; (4) a list of categories. Each anchor refers to the entity mention in the text, which can be linked to its Wikipedia page. Each category is a descriptive concept (definition) of an entity. For example, the entity "*Agatha Christie*" is defined as "*Murder–suicide in fiction*",

*"English women dramatists"* and *"British novels adapted into films"*. Therefore, we can naturally connect the entity mentions in contexts and descriptive concepts in Wikipedia pages through anchors.

To collect more entity types, we also link the entity to Wikidata (Vrandečić and Krötzsch, 2014) via its Wikipedia id. For each entity, We use the `instance of` and `subclass of` property values as its corresponding entity types (Chen et al., 2022). In this way, we automatically construct a large-scale pre-training corpus called Wikipair80M, containing 80M context-type pairs, such as ("*And Then There Were None is a novel by Christie.*", `British detective fiction writers`). We also obtain a collection of 120K entity types which can serve as the foundation for a much larger vocabulary.

### 3.2 Open-Vocabulary Named Entity Recognition Fine-tuning

In the open-vocabulary named entity recognition fine-tuning stage, we first fine-tune the recognizer on base types $\mathcal{T}_B$ with human annotations. Then, we expand the type query embeddings to include novel types $\mathcal{T}_N$ for OVNER at inference time.

**Architecture** Dual-Encoder is computationally efficient for pre-training because it encodes the input of context and type independently. However, Dual-Encoder ignores the interactions between context and type. Li et al. (2020) and Aly et al. (2021) formulate NER as an MRC-style task and encode the input context and entity types jointly. At inference time, every type must be concatenated with the input context and fed into the model through a forward pass. Due to the high time complexity of inference, this method may not work for hundreds of types in real-time systems.

We employ Cross-Encoder architecture to capture the rich interactions between context and all candidate types efficiently and effectively. Recently, cross-attention modules have been successfully used to learn cross-modal interactions in multi-modal tasks (Wei et al., 2020; Kim et al., 2021; Li et al., 2021b), motivating us to leverage cross-attention modules for exploring context-type interactions. As shown in Figure 2 (c), Cross-Encoder $E_{\text{cross}}$ is based on Context-Encoder $E_{\text{ctxt}}$ and Type-Encoder $E_{\text{type}}$. Cross-Encoder is composed of a stack of $k = 3$ cross-attention layers and feed-forward networks. Cross-Encoder takes the context representations $[\mathbf{h}_{c,1}, \mathbf{h}_{c,2}, \ldots, \mathbf{h}_{c,n}]$ and

all the type representations $[\mathbf{h}_{t,1}, \mathbf{h}_{t,1}, \ldots, \mathbf{h}_{t,|\mathcal{T}_B|}]$ encoded by Dual-Encoder as input:

$$
\begin{aligned}
&\left[\mathbf{h}_{c,1}^{'}, \ldots, \mathbf{h}_{c,n}^{'}, \mathbf{h}_{t,1}^{'}, \ldots, \mathbf{h}_{t,|\mathcal{T}_B|}^{'}\right] \\
&= E_{\text{cross}}\left(\left[\mathbf{h}_{c,1}, \ldots, \mathbf{h}_{c,n}, \mathbf{h}_{t,1}, \ldots, \mathbf{h}_{t,|\mathcal{T}_B|}\right]\right).
\end{aligned} \tag{9}
$$

**Scoring** The similarity score between the token in context $c$ and the type $t$ is computed as:

$$
\text{score}\left(\mathbf{h}_c, \mathbf{h}_t\right) = \text{sigmoid}\left(\text{sim}\left(\mathbf{h}_c, \mathbf{h}_t\right)/\tau\right). \tag{10}
$$

**Optimization** Since `None` type is hard to describe using natural language, we ignore the `None` type and formulate NER as a binary classification problem. Then, we fine-tune the model using the following binary cross-entropy (BCE) loss:

$$
\begin{aligned}
\mathcal{L}_{bce} = \frac{1}{n\,|\mathcal{T}_B|} \sum_{i=1}^{n} \sum_{j=1}^{|\mathcal{T}_B|} & y_{c,i}^{j} \log p_{c,i}^{j} \\
&+ \left(1 - y_{c,i}^{j}\right) \log\left(1 - p_{c,i}^{j}\right),
\end{aligned} \tag{11}
$$

where $p_{c,i}^{j} = \text{score}\left(\mathbf{h}_{c,i}^{'}, \mathbf{h}_{t,j}^{'}\right)$ denotes the probability of the $i$-th token in context $c$ belongs to $j$-th type, $y_{c,i}^{j} \in \{0, 1\}$ denotes the ground truth whether the $i$-th token in context $c$ belongs to the $j$-th type.

**Inference** At inference time, we predict the type $\hat{t}_{c,i}$ of the $i$-th token as follows:

$$
\hat{t}_{c,i} = \begin{cases} \text{argmax}(\mathbf{p}_{c,i}), & \text{if } \max(\mathbf{p}_{c,i}) \geq \delta \\ \text{None}, & \text{otherwise}, \end{cases} \tag{12}
$$

where $\delta$ is a threshold hyper-parameter.

## 4 Experiments

### 4.1 Evaluation Benchmarks and Metrics

**Evaluation Benchmark** So far, there is no benchmark for evaluating OVNER models. Therefore, we adapt OntoNotes 5.0 (Pradhan et al., 2013) and FEW-NERD (Ding et al., 2021) datasets into the OVNER setting. OntoNotes 5.0 is a widely used NER dataset containing eleven entity types and seven value types. Few-NERD is the largest few-shot NER dataset, which contains INTRA and INTER two sub-settings with eight coarse-grained and 66 fine-grained types. Following the previous zero-shot NER method (Aly et al., 2021), we construct three challenging OVNER benchmarks, namely OV-OntoNotes, OV-NERD-INTRA, and OV-NERD-INTER. Within these benchmarks, we

| Method | PLM | OV-OntoNotes | | | OV-NERD-INTRA | | | OV-NERD-INTER | | |
|---|---|---|---|---|---|---|---|---|---|---|
| | | Base | Novel | All | Base* | Novel | All | Base | Novel | All |
| | | | | | Baseline Methods | | | | | |
| MRC | BERT-base-SQuAD | 80.5 | 11.7 | 36.7 | - | 19.4 | 19.4 | 66.2 | 10.1 | 40.7 |
| MRC | BERT-large-SQuAD | 81.3 | 14.4 | 38.7 | - | 21.3 | 21.3 | 67.5 | 11.5 | 42.0 |
| BEM | BERT-base-MNLI | 78.6 | 14.8 | 38.0 | - | 16.6 | 16.6 | 64.4 | 9.4 | 39.4 |
| BEM | BERT-large-MNLI | 79.8 | 17.2 | 40.0 | - | 18.7 | 18.7 | 65.2 | 10.3 | 40.3 |
| SMXM | BERT-base | 79.7 | 18.5 | 40.8 | - | 20.2 | 20.2 | 66.9 | 11.2 | 41.6 |
| SMXM | BERT-large | 80.8 | 21.8 | 43.3 | - | 23.4 | 23.4 | 68.0 | 13.4 | 43.2 |
| | | | | | Our Method | | | | | |
| CACAO | BERT-base | **82.7** | **23.6** | **45.1** | - | **33.1** | **33.1** | **74.2** | **22.9** | **50.9** |

Table 1: Experimental results (macro-averaged F1-score, %) of different methods on OV-OntoNotes, OV-NERD-INTRA and OV-NERD-INTER benchmarks. Bold denotes best results. * indicates that the test set has no base types, since the entity types are mutually disjoint in the training set and test set of OV-NERD-INTRA.

| Method | OV-OntoNotes | | | OV-NERD-INTRA | | | OV-NERD-INTER | | |
|---|---|---|---|---|---|---|---|---|---|
| | Base | Novel | All | Base | Novel | All | Base | Novel | All |
| CACAO | **82.7** | 23.6 | 45.1 | - | **33.1** | **33.1** | **74.2** | 22.9 | **50.9** |
| *w/o* pre-training | 82.3 | 0.0 | 29.9 | - | 14.5 | 14.5 | 54.9 | 6.8 | 33.0 |
| *w/o* in-instance | 81.9 | 21.6 | 43.5 | - | 32.1 | 32.1 | 74.1 | 19.7 | 49.4 |
| *w/o* in-batch | 81.6 | 20.7 | 42.9 | - | 31.7 | 31.7 | 72.9 | 20.1 | 48.9 |
| *w/o* cross-encoder | 80.7 | **25.4** | **45.5** | - | 30.4 | 30.4 | 73.0 | 19.4 | 48.6 |
| *w/* self-attention | 82.5 | 21.0 | 43.4 | - | 31.3 | 31.3 | 74.0 | 22.3 | 50.5 |

Table 2: Ablation study of CACAO on OV-OntoNotes, OV-NERD-INTRA and OV-NERD-INTER benchmarks.

consider those types seen in the training set as base types. The test set may contain both novel types and base types, depending on the specific benchmark setting. The statistics and type splits of the three benchmarks are shown in Appendix A.

**Evaluation Metric**   We consider *macro-averaged F1-score* to evaluate the performance due to the imbalance in annotated samples per type. We experiment with three random seeds and report the performance of base types, novel types and all types.

### 4.2   Baseline Methods

We compare the proposed approach CACAO with the following methods:

(1) **MRC**: Li et al. (2020) formulate the NER task as a machine reading comprehension task and construct natural language queries for entity types. For example, extracting entities with the `Person` type is transformed as extracting answer spans to the question "*which person is mentioned in the text?*". We adopt BERT-base and BERT-large fine-tuned on SQuAD (Rajpurkar et al., 2016) dataset.

(2) **BEM**: Yin et al. (2019) propose to treat zero-shot text classification as a textual entailment task and classify whether a class description is entailed by the text. Following Aly et al. (2021), we adjust BEM for OVNER. We design a template "*This text is about ...*" to convert each entity type into a hypothesis, then treat the target mention in context as the premise, and ask the BEM model whether the target mention can entail an entity type. We adopt BERT-base and BERT-large fine-tuned on MNLI (Williams et al., 2018) dataset.

(3) **SMXM**: Aly et al. (2021) present the first approach for zero-shot named entity recognition. SMXM uses the cross-attention encoder to model the sentence and the entity type descriptions and explores three approaches to modeling `None` type. We adopt BERT-base and BERT-large for SMXM.

All three baseline methods and our CACAO need to be fine-tuned on base types. The implementation details of CACAO are shown in Appendix B.

### 4.3   Overall Results

Table 1 shows the results of different methods on OV-OntoNotes, OV-NERD-INTRA and OV-NERD-INTER benchmarks. We note the following

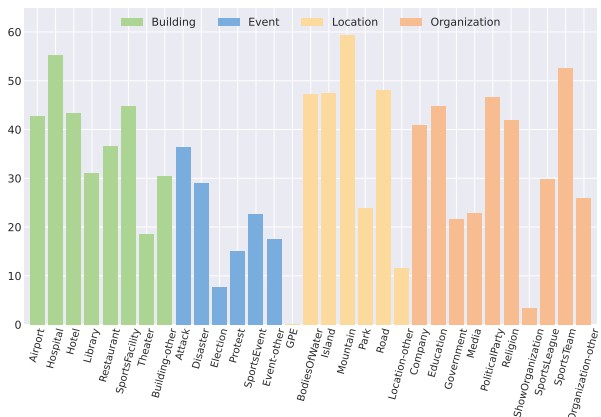

Figure 3: F1-scores of different novel entity types on OV-NERD-INTRA.

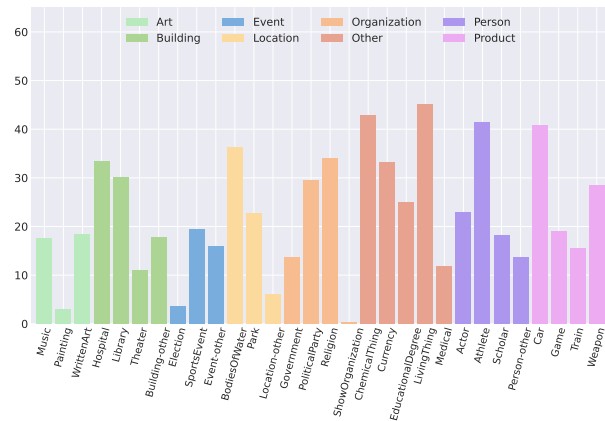

Figure 4: F1-scores of different novel entity types on OV-NERD-INTER.

| Pooler red $(\cdot)$ | OV-NERD-INTRA | | | OV-NERD-INTER | | |
|---|---|---|---|---|---|---|
| | Base | Novel | All | Base | Novel | All |
| [CLS] | - | 27.3 | 27.3 | 70.4 | 18.8 | 46.9 |
| *name* | - | 22.6 | 22.6 | 67.5 | 13.6 | 43.0 |
| **mean** | - | **31.1** | **31.1** | **74.2** | **22.9** | **50.9** |

Table 3: Effects of different pooling functions.

| Desc | OV-NERD-INTRA | | | OV-NERD-INTER | | |
|---|---|---|---|---|---|---|
| | Base | Novel | All | Base | Novel | All |
| Wiki | - | 31.1 | 31.1 | **74.2** | **22.9** | **50.9** |
| WordNet | - | 23.6 | 23.6 | 66.4 | 12.3 | 41.8 |
| GPT-3.5 | - | **31.8** | **31.8** | 71.7 | 19.7 | 48.1 |

Table 4: Effects of different type descriptions.

key observations throughout our experiments:

(1) Our method outperforms all the baseline methods by a large margin and achieves new state-of-the-art performance on three benchmarks. More specially, compared with the previous state-of-the-art model SMXM, our method achieves 1.8%, 9.7%, and 9.5% improvements of F1-score of novel types on OV-OntoNotes, OV-NERD-INTRA and OV-NERD-INTER benchmarks, respectively. Experimental results demonstrate that our proposed CACAO is very effective for the open-vocabulary named entity recognition task.

(2) Our CACAO performs well not only on novel types but also on base types. It indicates that CACAO can learn a rich and complete context-type semantic space, and expand its vocabulary from large-scale pre-training. Furthermore, we also observe that the performance improvement of CACAO is more significant on OV-NERD-INTRA compared to OV-OntoNotes. This suggests CACAO has effective fine-grained entity type recognition ability.

(3) MRC or BEM design a template to convert each entity type name into a question or a hypothesis. Compared with them using type names, our CACAO and SMXM adopting entity type descriptions achieve better performance. The reason is that type descriptions can help the model understand the meaning of types and generalize to new types.

## 4.4 Ablation Results

To demonstrate the effectiveness of our proposed CACAO, we conduct ablation studies as follows: 1) *w/o* pre-training, which removes the entire pre-training stage; 2) *w/o* in-instance, which removes in-instance contrastive learning from the pre-training stage; 3) *w/o* in-batch, which removes in-batch contrastive learning from the pre-training stage; 4) *w/o* cross-encoder, which removes Cross-Encoder from the fine-tuning stage; 5) *w/* self-attention, which replaces the cross-attention layers with the self-attention layers in Cross-Encoder. From Table 2 we have the following observations:

(1) We observe a significant performance drop of 23.6%, 18.6%, and 16.1% in the F1-score of CACAO when dealing with novel types without pre-training. This highlights the effectiveness of alignment pre-training in improving generalization to unseen types. We also find that in-batch contrastive pre-training is more efficient than in-instance contrastive pre-training. The reason may be that in-instance contrastive pre-training will introduce false negative examples in an instance.

(2) When we remove Cross-Encoder, the performance drops significantly on OV-NERD-INTRA and OV-NERD-INTER, but improves on OV-OntoNotes. This phenomenon may be attributed to the fact that Few-NERD has much more entity

| Method | PLM | Politics | Science | Music | Literature | AI | Average |
|---|---|---|---|---|---|---|---|
| LST-NER[†] | BERT-base | 70.44 | 66.83 | 72.08 | 67.12 | 60.32 | 67.36 |
| LANER[†] | BERT-base | 71.65 | 69.29 | 73.07 | 67.98 | 61.72 | 68.74 |
| CP-NER[†] | T5-base | 73.41 | 74.65 | 78.08 | 70.84 | 64.53 | 72.30 |
| LST-NER *w/* DAPT[†] | BERT-base | 73.25 | 70.07 | 76.83 | 70.76 | 63.28 | 70.84 |
| LANER *w/* DAPT[†] | BERT-base | 74.06 | 71.83 | 78.78 | 71.11 | 65.79 | 72.31 |
| CP-NER *w/* DAPT[†] | T5-base | 76.35 | 76.83 | 80.28 | 72.17 | 66.39 | 74.40 |
| CACAO (Ours) | BERT-base | **81.26** | **77.78** | **82.60** | **75.73** | **68.95** | **77.26** |

Table 5: Experimental results (micro-averaged F1-score, %) of different methods on CrossNER dataset. † indicates the results taken from the original paper (Chen et al., 2023b).

types than OntoNotes. Cross-Encoder works well with a larger number of types, capturing the intricate interactions between context and types. In addition, we find that the cross-attention mechanism performs better than the self-attention mechanism.

### 4.5 Discussion and Analysis

**Performance of Different Entity Types** We analyze the performance of different entity types on OV-NERD-INTRA and OV-NERD-INTER. As illustrated in Figure 3 and 4, we find that CACAO achieves a good performance on `Person` and `Building` types and yields a bad performance on `Event` type. Moreover, CACAO can barely recognize `GPE`, `ShowOrganization` and `Election` types. A possible explanation is these types are difficult to define and describe in a single sentence. For instance, we describe `GPE` as "*geographical/social/political entity, including countries, cities, states*". Therefore, a good type description is crucial for recognizing novel types.

**Impact of Entity Type Representations** We experiment with three $\mathrm{red}\left(\cdot\right)$ pooling functions to represent the entity types: 1) `[CLS]`: using the output vector of the `[CLS]` token; 2) *name*: using the output vector of the *name* token; 3) **mean**: computing the average of all output vectors. As illustrated in Table 3, **mean** can provide more comprehensive information about entity types, leading to better performance than `[CLS]` and *name*.

**Impact of Entity Type Descriptions** We also analyze the effect of entity type descriptions by conducting experiments with three different type descriptions: 1) Wiki: the most relevant sentence describing the type that we carefully selected on Wikipedia; 2) WordNet: the type definition in WordNet (Miller, 1995); 3) GPT-3.5: the type

description generated by `text-davinci-003`. Table 4 demonstrates that Wiki, with its well-crafted descriptions, can serve as a strong baseline compared to WordNet. Meanwhile, using GPT-3.5 to automatically generate descriptions can also achieve good performance and reduce human labor. Please refer Appendix D for entity descriptions.

### 4.6 Transfer Ability

We also demonstrate the transfer ability of our CACAO in cross-domain NER. We conduct experiments on the CrossNER dataset (Liu et al., 2021). Following the previous studies (Zheng et al., 2022; Hu et al., 2022; Chen et al., 2023b), we adopt CoNLL 2003 (Sang and De Meulder, 2003) as the source domain which contains four types: `PER`, `LOC`, `ORG`, and `MISC`. Then, we use CrossNER as the target domain which contains five separate domains: politics, natural science, music, literature and AI. Table 7 presents the detailed statistics of these datasets. To evaluate the effectiveness of CACAO, we compare it with several cross-domain NER baselines, including:

(1) **DAPT**: Liu et al. (2021) propose domain-adaptive pre-training (DAPT), which leverages a large domain-related corpus to pre-train the model, similar to the first-stage pre-training of our method.

(2) **LST-NER**: Zheng et al. (2022) build label graphs in both the source and target label spaces based on BERT-base for cross-domain NER.

(3) **LANER**: Hu et al. (2022) propose an autoregressive cross-domain NER framework based on BERT-base to improve label information transfer.

(4) **CP-NER**: Chen et al. (2023b) introduce collaborative domain-prefix tuning for cross-domain NER based on T5-base (Raffel et al., 2020), which obtains the state-of-the-art performance on the CrossNER dataset.

To make a fair comparison, we first train on the source domain with sufficient data, and then adapt to the target domain with a very small amount of data. As shown in Table 5, CACAO can consistently obtain better performance than the previous state-of-the-art method CP-NER *w/* DAPT across all target domains, with an average 2.86% F1-score improvement. Experimental results demonstrate the effectiveness of our OVNER method under cross-domain NER scenarios. We think there are several reasons for the remarkable results:

(1) CACAO can learn a complete context-type semantic space, and expand its vocabulary from large-scale pre-training. Pre-training is essential to improve the domain generalization of our model.

(2) CACAO leverages text descriptions to represent entity types, which can contain more type semantic information. We guess entity descriptions can better realize the transfer of knowledge from the source domain to the target domain.

## 5 Related Work

### 5.1 Supervised Named Entity Recognition

Named entity recognition (NER) is a long-standing study in natural language processing. Traditional approaches for NER are based on sequence labeling (Ju et al., 2018; Wang et al., 2020), assigning a single tag to each word in a sentence. To handle overlapping entities in nested NER, span-based methods (Luan et al., 2019; Fu et al., 2021; Li et al., 2021a; Shen et al., 2021; Chen et al., 2023a) first extract all candidate spans and then classify these spans. Besides, Li et al. (2020) and Shen et al. (2022) propose to reformulate the NER task as a machine reading comprehension (MRC) task. In addition, some methods (Yan et al., 2021; Ma et al., 2022; Chen et al., 2022; Lu et al., 2022b) attempt to formulate NER as an autoregressive sequence generation task and achieve promising performance. Das et al. (2022) propose to solve few-shot NER via contrastive learning. Recently, Zhang et al. (2023) present a bi-encoder framework, which separately maps text and entity types into the same vector space. These methods have achieved high performance in the supervised setting, yet they all require costly annotations. Hence, it is hard for them to handle novel-emerging types.

### 5.2 Zero-Shot Named Entity Recognition

Zero-shot named entity recognition (ZSNER) aims to recognize entities in novel types without any training samples. Aly et al. (2021) present the first approach for zero-shot named entity recognition and leverage the entity type descriptions to transfer knowledge from observed to unseen classes. Nevertheless, existing ZSNER methods are still far from practical performance due to their unnecessarily harsh constraint. In this paper, we consider a more realistic setting, in which not only the annotated samples of base types but also a large number of low-cost context-type pairs are available.

### 5.3 Open-Vocabulary Object Detection

Open-vocabulary object detection (OVD) (Zareian et al., 2021; Zang et al., 2022; Li et al., 2022b; Gu et al., 2022; Zhou et al., 2022; Xu et al., 2022; Kuo et al., 2022), which is concerned with the problem of detecting novel objects guided by natural language, has gained increasing attention from the community. Zareian et al. (2021) first propose the open-vocabulary object detection task to bridge the performance gap between zero-shot learning and supervised learning. They first pre-train the model on image-caption pairs and then fine-tune the model on downstream detection datasets. There are some studies (Li et al., 2021c, 2022a,c; Jiao et al., 2022; Lu et al., 2022a) in the NLP community that use indirect supervision to alleviate the data scarcity issue. Inspired by these outstanding works, we propose to solve open-vocabulary named entity recognition with context-type semantic alignment, further improving the zero-shot performance on novel-emerging entity types.

## 6 Conclusion

In this paper, we consider a more realistic and challenging setting called open-vocabulary named entity recognition (OVNER) to imitate human-level ability. We formulate OVNER as a semantic matching task and propose a two-stage method called **C**ontext-Type Sem**A**nti**C** **A**lignment and Fusi**O**n (CACAO). In the pre-training stage, we first construct a corpus containing 80M context-type pairs, then pre-train Dual-Encoder through natural supervision for context-type semantic alignment. In the fine-tuning stage, we fine-tune Cross-Encoder on base types with human supervision for context-type semantic fusion, so that the model can recognize novel types by their descriptions. Experimental results show that our method outperforms the state-of-the-art methods on three challenging benchmarks.

## Limitations

Although our approach has worked well on open-vocabulary named entity recognition task, there are still some limitations to be resolved in the future: (1) The core idea of OVNER is bridging the gap between zero-shot NER and supervised NER by learning from natural supervision. In this paper, we pre-train the recognizer on large-scale context-type pairs as natural supervision. Despite its effectiveness, pre-training requires significant amounts of computational resources. In the future, We will explore a general way to leverage natural supervision; (2) In this paper, we solely evaluate our method CACAO on several OVNER benchmarks. In fact, our method can be adapted to solve all kinds of entity-related information extraction tasks, including named entity recognition, entity typing, entity clustering, and entity linking. We leave it as an open question for future work. (3) Our method requires the provision of textual descriptions to represent entity types. Although we think it may be an easier way to provide or collect entity descriptions for each type. We believe that alleviating the limitation of type descriptions could be a further extension of our work in the future.

## Ethics Statement

To ensure the reproducibility of our paper, we will release all source codes, large-scale pre-training corpus, benchmark datasets, and all trained checkpoints upon the acceptance of this paper.

## Acknowledgements

This work is supported by the National Key Research and Development Program of China (No. 2020AAA0106400), the National Natural Science Foundation of China (No. 6197621162176257 ). This work is also supported by the Strategic Priority Research Program of Chinese Academy of Sciences (Grant No.XDA27020100 ), the Youth Innovation Promotion Association CAS, and Yunnan Provincial Major Science and Technology Special Plan Projects (No.202202AD080004).

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

## A Benchmark Details

Table 6 presents the statistics of OV-OntoNotes, OV-NERD-INTRA and OV-NERD-INTER benchmarks. Table 7 presents the statistics of CoNLL 2003 and CrossNER datasets. Additionally, we also provide detailed type splits in Table 8, 9 and 10.

## B Implementation Details

Our implementation is based on HuggingFace's Transformers library and PyTorch. We initialize Context-Encoder and Type-Encoder with pretrained BERT-base. Then, we initialize Cross-Encoder with the last three layers of pre-trained BERT-base. We use the Adam algorithm (Kingma and Ba, 2015) to optimize model parameters. The learning rate is initialized as 1e-5 and 5e-6 with a linear decay for pre-training and fine-tuning stages. The batch size is set to 100 and 30 for pre-training and fine-tuning stages. We also keep a dropout of 0.1 and a weight decay of 0.01. The hyperparameter temperature $\tau$ is set to 0.05, $\mu_1$ is set to 0.5, $\mu_2$ is set to 0.3, mask probability $p$ is set to 0.5. The threshold $\delta$ is essential for inference. In general, using the default $\delta = 0.5$ can yield good performance. To achieve optimal performance, we tune the model on the development set with $\delta = \{0.4, 0.5, 0.6\}$. All experiments are conducted with eight NVIDIA GeForce RTX A6000 GPUs.

## C Case Study

As depicted in Table 11, we perform a case study to validate the effectiveness of our method on the OV-NERD-INTRA benchmark. The results demonstrate that our approach excels in accurately classifying entities. However, it faces challenges in precisely determining entity boundaries.

## D Type Descriptions

We present detailed entity descriptions of the OntoNotes and FEW-NERD datasets used in this paper shown in Table 12 and 13.

| Statistics | OV-OntoNotes | | | OV-NERD-INTRA | | | OV-NERD-INTER | | |
|---|---|---|---|---|---|---|---|---|---|
| | Train | Valid | Test | Train | Valid | Test | Train | Valid | Test |
| # Overall Types | 4 | 11 | 11 | 35 | 14 | 31 | 36 | 49 | 66 |
| # Novel Types | 0 | 7 | 7 | 0 | 14 | 31 | 0 | 13 | 30 |

Table 6: Statistics of OV-OntoNotes, OV-NERD-INTRA and OV-NERD-INTER benchmarks.

| Statistics | CoNLL 2003 | CrossNER | | | | |
|---|---|---|---|---|---|---|
| Domain | News | Politics | Natural Science | Music | Literature | Artificial Intelligence |
| # Train | 15.0k | 200 | 200 | 100 | 100 | 100 |
| # Test | 3.7k | 651 | 543 | 456 | 416 | 431 |
| # Entity Types | 4 | 10 | 17 | 13 | 11 | 12 |

Table 7: Statistics of CoNLL 2003 and CrossNER datasets.

| Split | Types |
|---|---|
| Train | PERSON, GPE, ORG, DATE |
| Dev | PERSON, GPE, ORG, DATE, FAC, LOC, TIME, QUANTITY, CARDINAL, WORK_OF_ART, LANGUAGE |
| Test | PERSON, GPE, ORG, DATE, NORP, MONEY, ORDINAL, PERCENT, EVENT, PRODUCT, LAW |

Table 8: Type split of OV-OntoNotes. Blue denotes the base type, red denotes the novel type.

| Split | Types |
|---|---|
| Train | product-weapon, product-train, product-software, product-ship, product-other, product-game, product-food, product-car, product-airplane, person-soldier, person-scholar, person-politician, person-other, person-director, person-athlete, person-artist/author, person-actor, other-medical, other-livingthing, other-law, other-language, other-god, other-educationaldegree, other-disease, other-currency, other-chemicalthing, other-biologything, other-award, other-astronomything, art-writtenart, art-painting, art-other, art-music, art-film, art-broadcastprogram |
| Dev | event-sportsevent, event-protest, event-other, event-election, event-disaster, event-attack/battle/war/militaryconflict, building-theater, building-sportsfacility, building-restaurant, building-other, building-library, building-hotel, building-hospital, building-airport |
| Test | building-airport, building-hospital, building-hotel, building-library, building-other, building-restaurant, building-sportsfacility, building-theater, event-attack/battle/war/militaryconflict, event-disaster, event-election, event-other, event-protest, event-sportsevent, location-GPE, location-bodiesofwater, location-island, location-mountain, location-other, location-park, location-road/railway/highway/transit, organization-company, organization-education, organization-government/governmentagency, organization-media/newspaper, organization-other, organization-politicalparty, organization-religion, organization-showorganization, organization-sportsleague, organization-sportsteam |

Table 9: Type split of OV-NERD-INTRA. Blue denotes the base type, red denotes the novel type.

| Split | Types |
|---|---|
| Train | art-broadcastprogram, art-film, art-other, building-airport, building-hotel, building-restaurant, building-sportsfacility, event-attack/battle/war/militaryconflict, event-disaster, event-protest, location-GPE, location-island, location-mountain, location-road/railway/highway/transit, organization-company, organization-education, organization-media/newspaper, organization-other, organization-sportsleague, organization-sportsteam, other-astronomything, other-award, other-biologything, other-disease, other-god, other-language, other-law, person-artist/author, person-director, person-politician, person-soldier, product-airplane, product-food, product-other, product-ship, product-software |
| Dev | art-broadcastprogram, art-film, art-other, building-airport, building-hotel, building-restaurant, building-sportsfacility, event-attack/battle/war/militaryconflict, event-disaster, event-protest, location-GPE, location-island, location-mountain, location-road/railway/highway/transit, organization-company, organization-education, organization-media/newspaper, organization-other, organization-sportsleague, organization-sportsteam, other-astronomything, other-award, other-biologything, other-disease, other-god, other-language, other-law, person-artist/author, person-director, person-politician, person-soldier, product-airplane, product-food, product-other, product-ship, product-software, art-painting, building-library, building-other, event-other, location-park, organization-religion, organization-showorganization, other-chemicalthing, other-currency, person-other, person-scholar, product-game, product-train |
| Test | art-broadcastprogram, art-film, art-other, building-airport, building-hotel, building-restaurant, building-sportsfacility, event-attack/battle/war/militaryconflict, event-disaster, event-protest, location-GPE, location-island, location-mountain, location-road/railway/highway/transit, organization-company, organization-education, organization-media/newspaper, organization-other, organization-sportsleague, organization-sportsteam, other-astronomything, other-award, other-biologything, other-disease, other-god, other-language, other-law, person-artist/author, person-director, person-politician, person-soldier, product-airplane, product-food, product-other, product-ship, product-software, art-music, art-painting, art-writtenart, building-hospital, building-library, building-other, building-theater, event-election, event-other, event-sportsevent, location-bodiesofwater, location-other, location-park, organization-government/governmentagency, organization-politicalparty, organization-religion, organization-showorganization, other-chemicalthing, other-currency, other-educationaldegree, other-livingthing, other-medical, person-actor, person-athlete, person-other, person-scholar, product-car, product-game, product-train, product-weapon |

Table 10: Type split of OV-NERD-INTER. Blue denotes the base type, red denotes the novel type.

| Sentences with Gold Entities | Sentences with Predicted Entities |
| --- | --- |
| The [Beinecke Rare Book]Building-Library and [Manuscript Library]Building-Library at Yale University has an archive of his collected papers. | The [Beinecke Rare Book and Manuscript Library]Building-Library at Yale University has an archive of his collected papers. |
| [Walker Cirque]Location-Mountain is a prominent glacier-filled cirque at the west side of the terminus of [McCleary Glacier]Location-Other in [Cook Mountains]Location-Mountain. | [Walker Cirque]Location-Bodiesofwater is a prominent glacier-filled cirque at the west side of the terminus of [McCleary Glacier]Location-Other in [Cook Mountains]Location-Mountain. |
| The [Warriors]Organization-Sportsteam were knocked out in the quarter-finals of the 2015 [NRL]Organization-Sportsleague [Auckland Nines]Organization-Sportsteam by eventual runners up [Cronulla Sutherland Sharks]Organization-Sportsteam. | The [Warriors]Organization-Sportsteam were knocked out in the quarter-finals of the [2015 NRL Auckland Nines]Organization-Sportsleague by eventual runners up [Cronulla Sutherland Sharks]Organization-Sportsteam. |
| It stands above [Brothers Water]Location-Bodiesofwater and the [Ullswater–Ambleside road]Location-Road/Railway/Highway. | It stands above [Brothers Water]Location-Bodiesofwater and the [Ullswater–Ambleside]Location-Other [road]Location-Road/Railway/Highway. |
| The wreck was sold at [Thursday Island]Location-Island according to the [Sydney Morning Herald]Organization-Media/Newspaper on 21 June 1883. | The wreck was sold at [Thursday Island]Location-Island according to the [Sydney Morning Herald]Organization-Media/Newspaper on 21 June 1883. |

Table 11: Case Study on OV-NERD-INTRA. Blue denotes the entites with gold annotations, green denotes the entites with true predictions, and red denotes the entites with false predictions.

| Name | Description |
| --- | --- |
| PERSON | person name [SEP] people, including fictional. |
| NORP | nationality, other, religion, political [SEP] nationalities or religious or political groups. |
| FAC | facility [SEP] buildings, airports, highways, bridges, etc. |
| ORG | organization [SEP] companies, agencies, institutions, etc. |
| GPE | geographical/social/political entity [SEP] countries, cities, states. |
| LOC | location [SEP] non-GPE locations, mountain ranges, bodies of water. |
| PRODUCT | product [SEP] vehicles, weapons, foods, etc. (Not services). |
| DATE | date [SEP] absolute or relative dates or periods. |
| TIME | time [SEP] times smaller than a day. |
| PERCENT | percent [SEP] percentage (including "%"). |
| MONEY | money [SEP] monetary values, including unit. |
| QUANTITY | quantity [SEP] measurements, as of weight or distance. |
| ORDINAL | ordinal [SEP] first, second. |
| CARDINAL | cardinal [SEP] numerals that do not fall under another type. |
| EVENT | event [SEP] named hurricanes, battles, wars, sports events, etc. |
| WORK_OF_ART | work of art [SEP] titles of books, songs, etc. |
| LAW | law [SEP] named documents made into laws. |
| LANGUAGE | language [SEP] any named language, English, Chinese. |

Table 12: Entity descriptions of OntoNotes.

| Name | Description |
| --- | --- |
| product-weapon | weapon [SEP] a weapon, arm or armament is any implement or device that can be used to deter, threaten, inflict physical damage, harm, or kill. Weapons are used to increase the efficacy and efficiency of activities such as hunting, crime, law enforcement, self-defense, warfare, or suicide. |
| product-train | train [SEP] in rail transport, a train is a series of connected vehicles that run along a railway track and transport people or freight. Trains are typically pulled or pushed by locomotives, though some are self-propelled, such as multiple units. |
| product-software | software [SEP] at the lowest programming level, executable code consists of machine language instructions supported by an individual processor—typically a central processing unit (CPU) or a graphics processing unit (GPU). Machine language consists of groups of binary values signifying processor instructions that change the state of the computer from its preceding state. |
| product-ship | ship [SEP] a ship is a large watercraft that travels the world's oceans and other sufficiently deep waterways, carrying cargo or passengers, or in support of specialized missions, such as defense, research, and fishing. Ships are generally distinguished from boats, based on size, shape, load capacity, and purpose. |
| product-other | product [SEP] in marketing, a product is an object, or system, or service made available for consumer use as of the consumer demand; it is anything that can be offered to a market to satisfy the desire or need of a customer. |
| product-game | game [SEP] a game is a structured form of play, usually undertaken for entertainment or fun, and sometimes used as an educational tool. |
| product-food | food [SEP] food is any substance consumed by an organism for nutritional support. Food is usually of plant, animal, or fungal origin, and contains essential nutrients, such as carbohydrates, fats, proteins, vitamins, or minerals. |
| product-car | car [SEP] a car or automobile is a motor vehicle with wheels. Most definitions of cars say that they run primarily on roads, seat one to eight people, have four wheels, and mainly transport people instead of goods. |
| product-airplane | airplane [SEP] an airplane or aeroplane (informally plane) is a fixed-wing aircraft that is propelled forward by thrust from a jet engine, propeller, or rocket engine. Airplanes come in a variety of sizes, shapes, and wing configurations. The broad spectrum of uses for airplanes includes recreation, transportation of goods and people, military, and research. |
| person-soldier | soldier [SEP] a soldier is a person who is a member of an army. A soldier can be a conscripted or volunteer enlisted person, a non-commissioned officer, or an officer. |
| person-scholar | scholar [SEP] a scholar is a person who pursues academic and intellectual activities, particularly academics who apply their intellectualism into expertise in an area of study. A scholar can also be an academic, who works as a professor, teacher, or researcher at a university. |
| person-politician | politician [SEP] a politician is a person active in party politics, or a person holding or seeking an elected office in government. Politicians propose, support, reject and create laws that govern the land and by extension its people. Broadly speaking, a politician can be anyone who seeks to achieve political power in a government. |

| person-other | person [SEP] a person is a being that has certain capacities or attributes such as reason, morality, consciousness or self-consciousness, and being a part of a culturally established form of social relations such as kinship, ownership of property, or legal responsibility. |
| --- | --- |
| person-director | director [SEP] a film director controls a film's artistic and dramatic aspects and visualizes the screenplay (or script) while guiding the film crew and actors in the fulfilment of that vision. The director has a key role in choosing the cast members, production design and all the creative aspects of filmmaking. |
| person-athlete | athlete [SEP] an athlete (also sportsman or sportswoman) is a person who competes in one or more sports that involve physical strength, speed, or endurance. |
| person-artist/author | artist/author [SEP] an artist is a person engaged in an activity related to creating art, practicing the arts, or demonstrating an art. An author is the writer of a book, article, play, mostly written work. |
| person-actor | actor [SEP] an actor or actress is a person who portrays a character in a performance. The actor performs in the flesh in the traditional medium of the theatre or in modern media such as film, radio, and television. The actor's interpretation of a role—the art of acting—pertains to the role played, whether based on a real person or fictional character. |
| other-medical | medical [SEP] medicine is the science[1] and practice[2] of caring for a patient, managing the diagnosis, prognosis, prevention, treatment, palliation of their injury or disease, and promoting their health. Medicine encompasses a variety of health care practices evolved to maintain and restore health by the prevention and treatment of illness. |
| other-livingthing | living thing [SEP] various forms of life exist, such as plants, animals, fungi, protists, archaea, and bacteria. |
| other-law | law [SEP] law is a set of rules that are created and are enforceable by social or governmental institutions to regulate behavior, with its precise definition a matter of longstanding debate. It has been variously described as a science and as the art of justice. |
| other-language | language [SEP] language is a structured system of communication. The structure of a language is its grammar and the free components are its vocabulary. Languages are the primary means of communication of humans, and can be conveyed through spoken, sign, or written language. |
| other-god | god [SEP] in monotheistic thought, God is usually viewed as the supreme being, creator, and principal object of faith. God is usually conceived of as being omnipotent, omniscient, omnipresent, and omnibenevolent, as well as having an eternal and necessary existence. |
| other-educationaldegree | educational degree [SEP] an academic degree is a qualification awarded to students upon successful completion of a course of study in higher education, usually at a college or university. These institutions commonly offer degrees at various levels, usually including undergraduate degrees, master's, and doctorates, often alongside other academic certificates and professional degrees. |
| other-disease | disease [SEP] a disease is a particular abnormal condition that negatively affects the structure or function of all or part of an organism, and that is not immediately due to any external injury. Diseases are often known to be medical conditions that are associated with specific signs and symptoms. A disease may be caused by external factors such as pathogens or by internal dysfunctions. |

| | |
|---|---|
| other-currency | currency [SEP] a currency is a standardization of money in any form, in use or circulation as a medium of exchange, for example banknotes and coins. A more general definition is that a currency is a system of money in common use within a specific environment over time, especially for people in a nation state. |
| other-chemicalthing | chemical thing [SEP] chemistry is the scientific study of the properties and behavior of matter. It is a natural science that covers the elements that make up matter to the compounds composed of atoms, molecules and ions: their composition, structure, properties, behavior and the changes they undergo during a reaction with other substances. |
| other-biologything | biology thing [SEP] biology is the scientific study of life. It is a natural science with a broad scope but has several unifying themes that tie it together as a single, coherent field. For instance, all organisms are made up of cells that process hereditary information encoded in genes, which can be transmitted to future generations. Another major theme is evolution, which explains the unity and diversity of life. |
| other-award | award [SEP] an award, sometimes called a distinction, is something given to a recipient as a token of recognition of excellence in a certain field.[1][2] When the token is a medal, ribbon or other item designed for wearing, it is known as a decoration. |
| other-astronomything | astronomy thing [SEP] astronomy is a natural science that studies celestial objects and phenomena. It uses mathematics, physics, and chemistry in order to explain their origin and evolution. Objects of interest include planets, moons, stars, nebulae, galaxies, and comets. |
| organization-sportsteam | sports team [SEP] a sports team is a group of individuals who play sports (sports player),[1] usually team sports, on the same team. The number of players in the group depends on type of the sports requirements. |
| organization-sportsleague | sports league [SEP] a sports league is a group of sports teams or individual athletes that compete against each other and gain points in a specific sport. At its simplest, it may be a local group of amateur athletes who form teams among themselves and compete on weekends. |
| organization-showorganization | show organization [SEP] show, an artistic production, such as: Concert, Radio show, Talk show, Television show, Theatre production. |
| organization-religion | religion [SEP] religion is usually defined as a social-cultural system of designated behaviors and practices, morals, beliefs, worldviews, texts, sanctified places, prophecies, ethics, or organizations, that generally relates humanity to supernatural, transcendental, and spiritual elements. |
| organization-politicalparty | political party [SEP] a political party is an organization that coordinates candidates to compete in a particular country's elections. It is common for the members of a party to hold similar ideas about politics, and parties may promote specific ideological or policy goals. |
| organization-other | organization [SEP] an organization or organisation, is an entity—such as a company, an institution, or an association—comprising one or more people and having a particular purpose. |
| organization-media/newspaper | media/newspaper [SEP] the news media or news industry are forms of mass media that focus on delivering news to the general public or a target public. These include news agencies, print media (newspapers, news magazines), broadcast news (radio and television), and the internet (online newspapers etc.). |

| | |
|---|---|
| organization-governmentagency | government/governmentagency [SEP] a government or state agency, sometimes an appointed commission, is a permanent or semi-permanent organization in the machinery of government that is responsible for the oversight and administration of specific functions, such as an administration. |
| organization-education | education [SEP] an educational institution is a place where people of different ages gain an education, including preschools, childcare, primary-elementary schools, secondary-high schools, and universities. They provide a large variety of learning environments and learning spaces. |
| organization-company | company [SEP] a company, abbreviated as co., is a legal entity representing an association of people, whether natural, legal or a mixture of both, with a specific objective. |
| location-road/railway/transit | road/railway/transit [SEP] a road is a linear way for the conveyance of traffic that mostly has an improved surface for use by vehicles (motorized and non-motorized) and pedestrians. |
| location-park | park [SEP] a park is an area of natural, semi-natural or planted space set aside for human enjoyment and recreation or for the protection of wildlife or natural habitats. |
| location-other | location [SEP] in geography, a location or place are used to denote a region, country, city, town, village or a hamlet (point, line, or area) on Earth's surface or elsewhere, such as. |
| location-mountain | mountain [SEP] a mountain is an elevated portion of the Earth's crust, generally with steep sides that show significant exposed bedrock. |
| location-island | island [SEP] an island (or isle) is an isolated piece of habitat that is surrounded by a dramatically different habitat, such as water. |
| location-bodiesofwater | bodies of water [SEP] a body of water or waterbody (often spelled water body) is any significant accumulation of water on the surface of Earth or another planet. The term most often refers to oceans, seas, and lakes, but it includes smaller pools of water such as ponds, wetlands, or more rarely, puddles. |
| location-GPE | GPE [SEP] the FAO geopolitical ontology is an ontology developed by the Food and Agriculture Organization of the United Nations (FAO) to describe, manage and exchange data related to geopolitical entities such as countries, territories, regions and other similar areas. |
| event-sportsevent | sports event [SEP] a multi-sport event is an organized sporting event, often held over multiple days, featuring competition in many different sports among organized teams of athletes from (mostly) nation-states. The first major, modern, multi-sport event of international significance was the Olympic Games. |
| event-protest | protest [SEP] a protest (also called a demonstration, remonstration or remonstrance) is a public expression of objection, disapproval or dissent towards an idea or action, typically a political one. Protests can be thought of as acts of cooperation in which numerous people cooperate by attending, and share the potential costs and risks of doing so. |
| event-other | event [SEP] an occurrence; something that happens. |
| event-election | election [SEP] an election is a formal group decision-making process by which a population chooses an individual or multiple individuals to hold public office. |
| event-disaster | disaster [SEP] a disaster is a serious problem occurring over a short or long period of time that causes widespread human, material, economic or environmental loss which exceeds the ability of the affected community or society to cope using its own resources. |

| | |
|---|---|
| event-attack/militaryconflict | attack/battle/war/militaryconflict [SEP] war is an intense armed conflict between states, governments, societies, or paramilitary groups such as mercenaries, insurgents, and militias. It is generally characterized by extreme violence, destruction, and mortality, using regular or irregular military forces. |
| building-theater | theater [SEP] theatre or theater is a collaborative form of performing art that uses live performers, usually actors or actresses, to present the experience of a real or imagined event before a live audience in a specific place, often a stage. |
| building-sportsfacility | sports facility [SEP] a sports complex is a group of sports facilities. For example, there are track and field stadiums, football stadiums, baseball stadiums, swimming pools, and Indoor arenas. |
| building-restaurant | restaurant [SEP] a restaurant is a business that prepares and serves food and drinks to customers. |
| building-other | building [SEP] a building, or edifice, is an enclosed structure with a roof and walls standing more or less permanently in one place, such as a house or factory (although there's also portable buildings). |
| building-library | library [SEP] a library is a collection of materials, books or media that are accessible for use and not just for display purposes. |
| building-hotel | hotel [SEP] a hotel is an establishment that provides paid lodging on a short-term basis. |
| building-hospital | hospital [SEP] a hospital is a health care institution providing patient treatment with specialized health science and auxiliary healthcare staff and medical equipment. |
| building-airport | airport [SEP] an airport is an aerodrome with extended facilities, mostly for commercial air transport. |
| art-writtenart | written art [SEP] literature is any collection of written work, but it is also used more narrowly for writings specifically considered to be an art form, especially prose fiction, drama, and poetry. |
| art-painting | painting [SEP] painting is the practice of applying paint, pigment, color or other medium to a solid surface. Painting is an important form in the visual arts, bringing in elements such as drawing, composition, gesture , narration , and abstraction. |
| art-other | art [SEP] art is a diverse range of human activity, and resulting product, that involves creative or imaginative talent expressive of technical proficiency, beauty, emotional power, or conceptual ideas. |
| art-music | music [SEP] music is generally defined as the art of arranging sound to create some combination of form, harmony, melody, rhythm or otherwise expressive content. |
| art-film | file [SEP] a film – also called a movie, motion picture, moving picture, picture or photoplay – is a work of visual art that simulates experiences and otherwise communicates ideas, stories, perceptions, feelings, beauty, or atmosphere through the use of moving images. These images are generally accompanied by sound and, more rarely, other sensory stimulations. |
| art-broadcastprogram | broadcast program [SEP] broadcast programming is the practice of organizing or ordering (scheduling) of broadcast media shows, typically radio and television, in a daily, weekly, monthly, quarterly or season-long schedule. |

Table 13: Entity descriptions of FEW-NERD.