# OpenReview forum: "Alignment Precedes Fusion: Open-Vocabulary Named Entity Recognition as Context-Type Semantic Matching"
_EMNLP/2023/Conference — EMNLP 2023 Findings_

### Official Review · Reviewer_79yH · 2023-08-03

**Soundness:** 3

**Excitement:**

3: Ambivalent: It has merits (e.g., it reports state-of-the-art results, the idea is nice), but there are key weaknesses (e.g., it describes incremental work), and it can significantly benefit from another round of revision. However, I won't object to accepting it if my co-reviewers champion it.

**Missing References:**

Li B, Yin W, Chen M. Ultra-fine entity typing with indirect supervision from natural language inference[J]. Transactions of the Association for Computational Linguistics, 2022, 10: 607-622.

Li D, Hu B, Chen Q. Prompt-based Text Entailment for Low-Resource Named Entity Recognition[C]//Proceedings of the 29th International Conference on Computational Linguistics. 2022: 1896-1903.

Jiao Y, Li S, Xie Y, et al. Open-vocabulary argument role prediction for event extraction[J]. arXiv preprint arXiv:2211.01577, 2022.

**Paper Topic And Main Contributions:**

This paper is about Open-Vocabulary Named Entity Recognition (NER), which is the task of identifying and classifying named entities in text, where the set of entity types is not fixed in advance. The paper proposes a novel and scalable two-stage method called Context-Type SemAntiC Alignment and FusiOn (CACAO) to recognize entities in novel types by their textual names or descriptions. The method is formulated as a semantic matching task and pre-trained on 80M context-type pairs, making it easily accessible through natural supervision. The paper addresses the challenge of open-vocabulary named entity recognition and makes several contributions towards a solution, including proposing the CACAO method, evaluating it on multiple datasets, and releasing the pre-trained models and code for reproducibility. The paper also provides a thorough analysis of the proposed method and its limitations, as well as directions for future work.

**Questions For The Authors:**

 Could you provide some examples of cases where the proposed method fails or makes errors, and how the method could be improved to handle such cases?

**Reasons To Accept:**

1. The strengths of this paper include proposing a novel and scalable two-stage method for open-vocabulary named entity recognition , which is a challenging and interesting task in natural language processing.
2. The paper provides a thorough analysis of the proposed method and its limitations, as well as directions for future work.
3. The paper also evaluates the proposed method on multiple datasets and releases the pre-trained models and code for reproducibility.


**Reasons To Reject:**

1. The paper mainly focuses on the English language, and it is unclear how well the proposed method would generalize to other languages.
2. The paper does not provide a detailed analysis of the computational complexity and efficiency of the proposed method, which could limit its practical applicability in some scenarios.
3. The paper does not provide a detailed analysis of the impact of different hyperparameters on the performance of the proposed method, which could limit the understanding of the robustness of the method.
4. It is better to compare with other LLM-based NER methods, which seems to be more suitable for zero-shot and open-vocabulary.

**Reproducibility:**

3: Could reproduce the results with some difficulty. The settings of parameters are underspecified or subjectively determined; the training/evaluation data are not widely available.

**Reviewer Confidence:**

3: Pretty sure, but there's a chance I missed something. Although I have a good feel for this area in general, I did not carefully check the paper's details, e.g., the math, experimental design, or novelty.

---

> ### Author Rebuttal · Authors · 2023-08-29
>
> Thanks for your careful and insightful reviews. Your professional reviews offer us great advice towards writing a more comprehensive and competitive paper!
>
> > **Response to Rejection 1**: The paper mainly focuses on the English language, and it is unclear how well the proposed method would generalize to other languages.
>
> This is a good idea! **Our method can be easily extended to multilingual.** Since Wikipedia covers over 300 language editions, we can easily build the pre-train **multilingual** corpus by following the data collection method proposed in our paper. Besides, we can replace the original backbone model with a multilingual pre-trained model, such as mBERT and XLMR.
>
> > **Response to Rejection 2**: The paper does not provide a detailed analysis of the computational complexity and efficiency of the proposed method, which could limit its practical applicability in some scenarios.
>
> Thanks for your valuable advice! **Our method is designed for practical NER applications**. **We first analyze the computational complexity of our method**. Since the input sentences and type descriptions are independently encoded via Dual-Encoder, **we can encode all entity types in advance**. Then, we employ Cross-Encoder to capture the rich interactions between context and all candidate types efficiently and effectively. At the inference stage, our method only need to encode the input sentence once, and then fuse the context and all types once, so its computational complexity is $O(1)$. Instead, MRC-style methods can only extract entities of one type per inference, so the model needs to be run $N$ times, where $N$ is the number of entity types. Due to the high time complexity $O(N)$ of inference, MRC-style methods may not work for hundreds of types in real-time systems.
>
> **We use the FLOPs-per-token estimates to calculate the compute cost of our method**: $C_{CACAO}=l_{input}\times\theta_{BERT}+(l_{input}+N)\times\theta_{BERT}$, where where $l_{input}=200$ is the number of tokens per input, $\theta_{BERT}=110M$ is the parameter size of BERT.
>
> The compute cost of MRC-style methods is: $C_{MRC}=N\times(l_{input}+l_{description})\times\theta_{BERT}$, where $l_{description}=50$ is the number of tokens per type description. Our method is efficient when the number of entity types $N=100\ or\ 1000$ is large in practical applications. If the online system has extremely high requirements for speed, **our method can directly discard Cross-Encoder, and only adopt Dual-Encoder for inference**. We can also use Faiss to perform efficient similarity search.
>
> According to your suggestion, we will provide a detailed analysis of the computational complexity and efficiency in the revised version.
>
> > **Response to Rejection 3**: The paper does not provide a detailed analysis of the impact of different hyperparameters on the performance of the proposed method, which could limit the understanding of the robustness of the method.
>
> Thanks for your suggestion! We provide the detailed hyperparameters in Appendix B. In fact, **our method is robust to hyperparameters**. For example, the threshold $\delta$ is essential for inference. In general, using the default $\delta=0.5$ can yield good performance. To achieve optimal performance, we tune the model on the development set with $\delta = \{0.4, 0.5, 0.6\}$.
>
> > **Response to Rejection 4**: It is better to compare with other LLM-based NER methods, which seems to be more suitable for zero-shot and open-vocabulary.
>
> Thanks for your suggestion! Because several recent work [1] show that LLMs are not effective information extractors in general, given their unsatisfactory performance in most settings. Besides, different from the LLM with the high latency and budget requirements, our method is designed for practical NER systems with low compute cost and latency. So we did not compare our method with LLM-based NER methods. We will try your suggestion and compare with LLM-based NER methods in the revised version.
>
> [1] Large Language Model Is Not a Good Few-shot Information Extractor, but a Good Reranker for Hard Samples!, Ma et al. 2023.
>
> > **Response to Question 1**: Could you provide some examples of cases where the proposed method fails or makes errors, and how the method could be improved to handle such cases?
>
> We think your suggestion is well! In Section 4.5 (Page 7), we analyze the performance of different entity types on OV-NERD-INTRA and OV-NERD-INTER. We find that **our model does not perform well when the entity types are difficult to define and describe in a single sentence**, such as TIME, PERCENT, QUANTITY and GPE types. Besides, we also perform a case study to analyze where our proposed method fails or makes errors. As shown in Table 9 (Page 15), our approach excels in accurately classifying entities. However, **it faces challenges in precisely determining entity boundaries**.
>
> Therefore, we can improve our method from the following directions:
>
> - **Provide several entity demonstrations to model** to enhance original type descriptions. We find this to be effective for identifying those hard-to-describe types.
> - Design new pre-training objectives to **improve the model's ability to determine entity boundaries**.
> - Increase the pre-training corpus or **use the retrieval-augmented method** to expand the model's vocabulary, so that the model can better handle long-tail types.
>
> > **Missing References**
>
> We really appreciate your suggestions on the details of our paper writing! Based on your suggestions, we now add the discussions of missing references [2, 3, 4] in the related work section.
>
> [2] Ultra-fine entity typing with indirect supervision from natural language inference, Li et al. 2022.
>
> [3] Prompt-based Text Entailment for Low-Resource Named Entity Recognition, Li et al. 2022.
>
> [4] Open-vocabulary argument role prediction for event extraction, Jiao et al. 2022.

---

### Official Review · Reviewer_3cyn · 2023-08-05

**Soundness:** 4
**Typos Grammar Style And Presentation Improvements:** A

**Excitement:**

3: Ambivalent: It has merits (e.g., it reports state-of-the-art results, the idea is nice), but there are key weaknesses (e.g., it describes incremental work), and it can significantly benefit from another round of revision. However, I won't object to accepting it if my co-reviewers champion it.

**Paper Topic And Main Contributions:**

This paper tackles a realistic open-vocabulary NER problem, i.e., recognizing novel types of entities given only their surface names/descriptions and annotation of base types.
The paper formulates OVNER as a semantic matching task and proposes a two-stage method. It first pretrains dual encoders on distantly-annotated context-type pairs for alignment based on contrastive learning, then it finetunes the cross-encoder on base type supervision.


**Questions For The Authors:**

Question A: Continual learning NER requires sufficient annotated data for new type, while continual few-shot NER does not. Comparing OVNER with this paradigm makes the discussion more complete in the introduction (Line 064).

Question B: It seems that the performance of novel types remains largely lower than that of base types. Discussion about future efforts to improve the performance and error analysis are expected.

**Reasons To Accept:**

The paper is easy to follow and well-motivated.
Abundant experimental results verify the effectiveness of the approach.


**Reasons To Reject:**

The paper only compares with discriminative methods, it is expected to compare with generative methods or LLMs which might perform better in some cases.

**Reproducibility:**

4: Could mostly reproduce the results, but there may be some variation because of sample variance or minor variations in their interpretation of the protocol or method.

**Reviewer Confidence:**

3: Pretty sure, but there's a chance I missed something. Although I have a good feel for this area in general, I did not carefully check the paper's details, e.g., the math, experimental design, or novelty.

---

> ### Author Rebuttal · Authors · 2023-08-29
>
> Thanks for your careful and insightful reviews. Your professional reviews offer us great advice towards writing a more comprehensive and competitive paper!
>
> > **Response to Rejection 1**: The paper only compares with discriminative methods, it is expected to compare with generative methods or LLMs which might perform better in some cases.
>
> Thanks for your insightful advice! **Considering that our model adopts the discriminative method, and previous generative methods do not perform very well in recognizing novel types, we only compare with several discriminative methods**. According to your suggestion, **we compare our method with several generative methods under cross-domain settings**. We use CoNLL 2003 as the source domain which is labeled with four types: PER, LOC, ORG, and MISC. Then, we use CrossNER [1] as the target domain which contains five separate domains: politics, natural science, music, literature and AI. To evaluate the effectiveness of our method, we compare it with several generative methods, including:
>
> - BARTNER [2]: BARTNER uses the pre-trained BART model to generate entity spans, formulating the NER task as a sequence generation problem.
> - LightNER [3]: LightNER utilizes a pluggable prompting method for pre-trained generative model to improve NER performance in low-resource settings.
> - CP-NER [4]: CP-NER introduces collaborative domain-prefix tuning for cross-domain NER based on text-to-text generative PLMs, which obtains SOTA performance on the CrossNER benchmark. CP-NER also leverages a large domain-related corpus to pre-train the model (DAPT), similar to the first-stage pre-training of our method.
>
> To make a fair comparison, we first train on the source domain with sufficient data, and then adapt to the target domain with a very small amount of data. As shown in **Table 1**, **our method can consistently obtain better performance than SOTA method CP-NER + DAPT across all target domains, with an average 2.86% F1-score improvement**. Experimental results demonstrate the effectiveness of our OVNER method under cross-domain NER scenarios. We think there are several reasons for the surprising results:
>
> - Our method can learn a rich and complete context-type semantic space, and expand its vocabulary from large-scale pre-training. **Pre-training is essential to improve the domain generalization of our model**.
> - Our method uses entity descriptions to represent types, which can contain more type semantic information. We guess entity descriptions can better **realize the transfer of knowledge from source domain to target domain**.
>
> **Table 1:** The performance of generative methods and our CACAO on CrossNER.
>
> | Models             | Politics  |  Science  |   Music   | Literature |    AI     |  Average  |  Average  |
> | ------------------ | :-------: | :-------: | :-------: | :--------: | :-------: | :-------: | :-------: |
> | BARTNER            |   69.90   |   65.14   |   65.35   |   58.93    |   53.00   |   62.46   |   62.46   |
> | LightNER           |   72.78   |   66.74   |   72.28   |   65.17    |   35.82   |   62.56   |   62.56   |
> | CP-NER             |   73.41   |   74.65   |   78.08   |   70.84    |   64.53   |   72.30   |   72.30   |
> | CP-NER+DAPT        |   76.35   |   76.83   |   80.28   |   72.17    |   66.39   |   74.40   |   74.40   |
> | CACAO (Our Method) | **81.26** | **77.78** | **82.60** | **75.73**  | **68.95** | **77.26** | **77.26** |
>
> [1] CrossNER: Evaluating Cross-Domain Named Entity Recognition, Liu et al. 2021.
>
> [2] Cross-domain Named Entity Recognition via Graph Matching, Zheng et al. 2022.
>
> [3] A Label-Aware Autoregressive Framework for Cross-Domain NER, Hu et al. 2022.
>
> [4] One Model for All Domains: Collaborative Domain-Prefix Tuning for Cross-Domain NER, Chen et al. 2023.
>
> > **Response to Question 1**: Continual learning NER requires sufficient annotated data for new type, while continual few-shot NER does not. Comparing OVNER with this paradigm makes the discussion more complete in the introduction (Line 064).
>
> Yes, we agree with you that continual few-shot NER requires only a small amount of annotated data to learn to recognize novel types. Compared with the outstanding continual few-shot NER methods [5], our method has the following differences: **OVNER focuses on generalizing novel types without any annotation, aiming to train a once-and-for-all recognizer**. To make our paper clearer and more complete, we will compare OVNER with this paradigm in our revised version.
>
> [5] Few-Shot Class-Incremental Learning for Named Entity Recognition, Wang et al. 2022.
>
> > **Response to Question 2**: It seems that the performance of novel types remains largely lower than that of base types. Discussion about future efforts to improve the performance and error analysis are expected.
>
> Thanks for your insightful suggestion! We can observe a performance gap between base types and novel types, which demonstrates that OVNER is a challenging and unsolved task. In Section 4.5 (Page 7), we analyze the performance of different entity types on OV-NERD-INTRA and OV-NERD-INTER. We find that **our model does not perform well when the entity types are difficult to define and describe in a single sentence**, such as TIME, PERCENT, QUANTITY and GPE types. Besides, we also perform a case study to analyze our method. As shown in Table 9 (Page 15), our approach excels in accurately classifying entities. However, **it faces challenges in precisely determining entity boundaries**.
>
> Therefore, we can improve our method from the following directions:
>
> - **Provide several entity demonstrations to model** to enhance original type descriptions. We find this to be effective for identifying those hard-to-describe types.
> - Design new pre-training objectives to **improve the model's ability to determine entity boundaries**.
> - Increase the pre-training corpus or **use the retrieval-augmented method** to expand the model's vocabulary, so that the model can better handle long-tail types.
>
> > **Writing Comments**: Typos Grammar Style And Presentation Improvements
>
> We really appreciate your suggestions on the details of our paper writing! We will revise our paper according to your suggestion.

---

### Official Review · Reviewer_tqWV · 2023-08-05

**Soundness:** 3

**Excitement:**

4: Strong: This paper deepens the understanding of some phenomenon or lowers the barriers to an existing research direction.

**Paper Topic And Main Contributions:**

This work proposes a more challenging task called Open Vocabulary Named Entity Recognition(OVNER) in response to a real-world need. The authors started with the motivation that the current NER approach does not work well to discover novel entity types. The goal of the newly proposed OVNER is to automatically recognize new types when given a description beside the name. For OVNER, the work also proposed a framework called CACAO and experimentally validated its effectiveness.

**Reasons To Accept:**

1. This work proposes a new Setting in an attempt to address an important challenge for NER in recent years, which is how to discover new entity types. Acceptance of this work may be useful for subsequent research in the NLP community.
2. The work is relatively complete and somewhat sound. While proposing OVNER, the CACAO framework is proposed to address it. The design and results of the experiment can relatively support their claims.

**Reasons To Reject:**

1. The work does not bridge well to past related research work in the community. This work criticizes past related research directions. OVNER should also have certain limitations, such as the provision of descriptions being necessary.
2. Does OVNER cover Domain-Adapted NER scenarios? Does it include scenarios with large domain differences?

**Reproducibility:**

3: Could reproduce the results with some difficulty. The settings of parameters are underspecified or subjectively determined; the training/evaluation data are not widely available.

**Reviewer Confidence:**

3: Pretty sure, but there's a chance I missed something. Although I have a good feel for this area in general, I did not carefully check the paper's details, e.g., the math, experimental design, or novelty.

---

> ### Author Rebuttal · Authors · 2023-08-29
>
> Thanks for your careful and insightful reviews. Your professional reviews offer us great advice towards writing a more comprehensive and competitive paper!
>
> > **Response to Rejection 1**: The work does not bridge well to past related research work in the community. This work criticizes past related research directions. OVNER should also have certain limitations, such as the provision of descriptions being necessary.
>
> We are sorry for making this confusion! We talked about past related research work (including supervised methods, continual learning methods and zero-shot methods) in the introduction, with the aim of presenting the motivation for our method **"open-vocabulary named entity recognition"**. **We think all of the previous work in the community is outstanding and interesting, making solid contributions to the development of NER**. Our work is also based on their research and aims to enhance the NER model's ability to handle novel-emerging types during the inference stage. Our intent is to make a meaningful and practical contribution to the community's research. We will revise the introduction so that our work bridges better with past related outstanding work.
>
> Furthermore, we agree with you that our approach has certain limitations. Compared to annotating the training data for new types, we think it may be an easier way to provide or collect entity descriptions for each type. As shown in Table 4 (Page 7), we also explore using GPT 3.5 to automatically generate descriptions instead of collecting descriptions from Wikipedia. Experimental results show that GPT 3.5 can also achieve good performance and reduce human labor. **We believe that alleviating the limitation of type descriptions could be a further extension of our work in the future**.
>
> > **Response to Rejection 2**: Does OVNER cover Domain-Adapted NER scenarios? Does it include scenarios with large domain differences?
>
> Thanks for your valuable suggestion! **We think OVNER covers Domain-Adapted NER scenarios, especially those with large domain differences.** According to your suggestion, we conduct experiments on the Cross-NER benchmark [1], and **are pleasantly surprised to find that our method has flexible transfer ability and performs well in cross-domain NER scenarios**. Following the previous work [2, 3, 4], we use CoNLL 2003 as the source domain which is labeled with four types: PER, LOC, ORG, and MISC. Then, we use CrossNER as the target domain which contains five separate domain: politics, natural science, music, literature and AI. **Table 1** presents the detailed statistics of these datasets.
>
> **Table 1:** The statistics of CoNLL 2003 and CrossNER.
>
> | Datasets     | CoNLL 2003 | CrossNER |    CrossNER     | CrossNER |  CrossNER  |        CrossNER         |
> | ------------ | :--------: | :------: | :-------------: | :------: | :--------: | :---------------------: |
> | Domain       |    News    | Politics | Natural Science |  Music   | Literature | Artificial Intelligence |
> | #Train       |   15.0k    |   200    |       200       |   100    |    100     |           100           |
> | #Test        |    3.7k    |   651    |       543       |   456    |    416     |           431           |
> | #Entity Type |     4      |    10    |       17        |    13    |     11     |           12            |
>
> To evaluate the effectiveness of our method, we compare it with several cross-domain NER baselines, including:
> - DAPT [1]: DAPT refers as domain-adaptive pre-training, which leverages a large domain-related corpus to pre-train the model, similar to the first-stage pre-training of our method.
> - LST-NER [2]: LST-NER builds label graphs in both the source and target label spaces for cross-domain NER tasks.
> - LANER [3]: LANER proposes a novel autoregressive cross-domain NER framework to improve label information transfer.
> - CP-NER [4]: CP-NER introduces collaborative domain-prefix tuning for cross-domain NER based on text-to-text generative PLMs, which obtains SOTA performance on the CrossNER benchmark.
>
> To make a fair comparison, we first train on the source domain with sufficient data, and then adapt to the target domain with a very small amount of data. As shown in **Table 2**, **our method can consistently obtain better performance than SOTA method CP-NER + DAPT across all target domains, with an average 2.86% F1-score improvement**. Experimental results demonstrate the effectiveness of our OVNER method under cross-domain NER scenarios. We think there are several reasons for the surprising results:
>
> - Our method can learn a rich and complete context-type semantic space, and expand its vocabulary from large-scale pre-training. **Pre-training is essential to improve the domain generalization of our model**.
> - Our method uses entity descriptions to represent types, which can contain more type semantic information. We guess entity descriptions can better **realize the transfer of knowledge from source domain to target domain**.
>
> **Table 2:** The performance of baselines and our CACAO on CrossNER.
>
> | **Models**         | **Politics** | **Science** | **Music** | **Literature** |  **AI**   | **Average** |
> | ------------------ | :----------: | :---------: | :-------: | :------------: | :-------: | :---------: |
> | LST-NER            |    70.44     |    66.83    |   72.08   |     67.12      |   60.32   |    67.36    |
> | LANER              |    71.65     |    69.29    |   73.07   |     67.98      |   61.72   |    68.74    |
> | CP-NER             |    73.41     |    74.65    |   78.08   |     70.84      |   64.53   |    72.30    |
> | LST-NER+DAPT       |    73.25     |    70.07    |   76.83   |     70.76      |   63.28   |    70.84    |
> | LANER+DAPT         |    74.06     |    71.83    |   78.78   |     71.11      |   65.79   |    72.31    |
> | CP-NER+DAPT        |    76.35     |    76.83    |   80.28   |     72.17      |   66.39   |    74.40    |
> | CACAO (Our Method) |  **81.26**   |  **77.78**  | **82.60** |   **75.73**    | **68.95** |  **77.26**  |
>
> [1] CrossNER: Evaluating Cross-Domain Named Entity Recognition, Liu et al. 2021.
>
> [2] Cross-domain Named Entity Recognition via Graph Matching, Zheng et al. 2022.
>
> [3] A Label-Aware Autoregressive Framework for Cross-Domain NER, Hu et al. 2022.
>
> [4] One Model for All Domains: Collaborative Domain-Prefix Tuning for Cross-Domain NER, Chen et al. 2023.

---

### Meta-Review · Area_Chair_rQii · 2023-09-19

**Recommendation:** 3

**Metareview:**

Summary:
This paper addresses a challenging open-vocabulary Named Entity Recognition (NER) problem, which involves recognizing new entity types based solely on their surface names/descriptions and annotations of base types. The approach formulates OVNER as a semantic matching task and presents a two-stage method. Initially, it pretrains dual encoders on context-type pairs with distant annotations for alignment using contrastive learning. Subsequently, it fine-tunes the cross-encoder with supervision on base types.

Reason To Accept:
1.This paper is well-structured and provides clear motivation. It introduces a novel approach to address a significant challenge in NER, specifically the discovery of new entity types. The acceptance of this work could serve as a valuable contribution to future research in the NLP community.
2.The research is comprehensive and reasonably robust. The paper includes a thorough analysis of the proposed method and its limitations, along with suggestions for future research directions.
3.The paper's strengths lie in its proposal of a novel and scalable two-stage method for open-vocabulary named entity recognition, a challenging and intriguing task within natural language processing.
4.The approach is thoroughly validated through extensive experiments, with evaluations conducted across multiple datasets. Additionally, the paper promotes reproducibility by releasing pre-trained models and code.

Reason To Reject:
1.The paper lacks a clear connection to prior related research in the field and is critical of previous research directions. It is essential to acknowledge certain limitations in the proposed approach, such as the necessity of providing descriptions.
2.Some experiments need to be considered. 1) The paper only compares with discriminative methods, it is expected to compare with generative methods or LLMs which might perform better in some cases. 2) The paper primarily focuses on the English language, leaving uncertainty about the generalizability of the proposed method to other languages.
3.Some details and analysis are lack. 1) A comprehensive analysis of the computational complexity and efficiency of the proposed method is lacking, potentially limiting its practical applicability in real-world scenarios. 2) The paper does not delve into an in-depth examination of how different hyperparameters affect the method's performance, hindering our understanding of its robustness.

---

### Decision · Program_Chairs · 2023-10-07

**Decision:**

Accept-Findings

**Comment:**

Summary:
This paper addresses a challenging open-vocabulary Named Entity Recognition (NER) problem, which involves recognizing new entity types based solely on their surface names/descriptions and annotations of base types. The approach formulates OVNER as a semantic matching task and presents a two-stage method. Initially, it pretrains dual encoders on context-type pairs with distant annotations for alignment using contrastive learning. Subsequently, it fine-tunes the cross-encoder with supervision on base types.

Reason To Accept:
1.This paper is well-structured and provides clear motivation. It introduces a novel approach to address a significant challenge in NER, specifically the discovery of new entity types. The acceptance of this work could serve as a valuable contribution to future research in the NLP community.
2.The research is comprehensive and reasonably robust. The paper includes a thorough analysis of the proposed method and its limitations, along with suggestions for future research directions.
3.The paper's strengths lie in its proposal of a novel and scalable two-stage method for open-vocabulary named entity recognition, a challenging and intriguing task within natural language processing.
4.The approach is thoroughly validated through extensive experiments, with evaluations conducted across multiple datasets. Additionally, the paper promotes reproducibility by releasing pre-trained models and code.

Reason To Reject:
1.The paper lacks a clear connection to prior related research in the field and is critical of previous research directions. It is essential to acknowledge certain limitations in the proposed approach, such as the necessity of providing descriptions.
2.Some experiments need to be considered. 1) The paper only compares with discriminative methods, it is expected to compare with generative methods or LLMs which might perform better in some cases. 2) The paper primarily focuses on the English language, leaving uncertainty about the generalizability of the proposed method to other languages.
3.Some details and analysis are lack. 1) A comprehensive analysis of the computational complexity and efficiency of the proposed method is lacking, potentially limiting its practical applicability in real-world scenarios. 2) The paper does not delve into an in-depth examination of how different hyperparameters affect the method's performance, hindering our understanding of its robustness.